# An Archaea-specific *c*-type cytochrome maturation machinery is crucial for methanogenesis in *Methanosarcina acetivorans*

**Dinesh Gupta[1], Katie E Shalvarjian[2], Dipti D Nayak[1]\***

[1]Department of Molecular and Cell Biology, University of California, Berkeley, Berkeley, United States; [2]Department of Plant and Microbial Biology, University of California, Berkeley, Berkeley, United States

**Abstract** *c*-Type cytochromes (cyt *c*) are proteins that undergo post-translational modification to covalently bind heme, which allows them to facilitate redox reactions in electron transport chains across all domains of life. Genomic evidence suggests that cyt *c* are involved in electron transfer processes among the Archaea, especially in members that produce or consume the potent greenhouse gas methane. However, neither the maturation machinery for cyt *c* in Archaea nor their role in methane metabolism has ever been functionally characterized. Here, we have used CRISPR-Cas9 genome editing tools to map a distinct pathway for cyt *c* biogenesis in the model methanogenic archaeon *Methanosarcina acetivorans,* and have also identified substrate-specific functional roles for cyt *c* during methanogenesis. Although the cyt *c* maturation machinery from *M. acetivorans* is universally conserved in the Archaea, our evolutionary analyses indicate that different clades of Archaea acquired this machinery through multiple independent horizontal gene transfer events from different groups of Bacteria. Overall, we demonstrate the convergent evolution of a novel Archaea-specific cyt *c* maturation machinery and its physiological role during methanogenesis, a process which contributes substantially to global methane emissions.

## Editor's evaluation

Within this manuscript the authors set out to determine the biogenesis of c-type cytochromes in methane producing archaea called methanogens. Compared to the bacterical cytochrome c assembly system, genes like ccmD, ccmH and ccmI are not found in archaea that contain a functional cytochrome Cs. They show that the proteins encoded by the ccmABCEF genes of *Methanosarcina acetivorans* are both essential and sufficient for cyt c biogenesis. They also show the substrate specific role of the mmcA y cyt c *M. acetivorans*. The authors do this using a combination of genetic, molecular, and physiological and biochemical analyses. These data are of high interest of scientists working on bioenergetics and yt c biogenesis in archaea.

## Introduction

*C*-type cytochromes (cyt(s) *c*) are found across the tree of life, and are critical to electron transfer processes ranging from aerobic respiration in the mitochondrion of Eukaryotes to intra/extracellular electron transport in Bacteria and Archaea (*Richardson, 2000*; *Bertini et al., 2006*; *Shi et al., 2016*; *Gupta et al., 2020*). The production of a holo-cyt *c*, that is capable of electron transfer, requires complex post-translational processing to covalently attach iron-protoporphyrin IX (heme *b*) to an

**\*For correspondence:**
dnayak@berkeley.edu

**Competing interest:** The authors declare that no competing interests exist.

**eLife digest** Archaea are single-celled organisms that were discovered over half a century ago. Recently, there has been a renewed interest in these microbes because theyplay a key role in climate change by controlling greenhouse gas emissions, like methane. Indeed, methane-producing Archaea generate nearly 70% of the methane gas released into the atmosphere.

A group of proteins called c-type cytochromes are essential to energy generation in several methane-producing archaea. However, it is a mystery how Archaea assemble their c-type cytochromes. In fact, genomic studies suggest that Archaea are missing some of the c-type cytochrome assembly machinery that bacteria use. This has led scientists to suspect that Archaea have an alternate mechanism for building these essential components.

To solve this mystery, Gupta, Shalvarjian, and Nayak used CRISPR-Cas9 gene-editing tools to characterize which proteins are essential for c-type cytochrome production in *Methanosarcina acetivorans*, a species of Archaea that produces methane. These experiments showed that *M. acetivorans* discarded a few parts of the process used by bacteria to generate c-type cytochromes, streamlining the assembly of these proteins. By comparing the genes of different Archaeal species, Gupta, Shalvarjian and Nayak were able to determine that Archaea acquired the genes for producing c-type cytochromes from bacteria via horizontal gene transfer, a process in which genes move directly from one organism into another. The streamlining of the process took place later, as different Archaeal species evolved independently, but losing the same parts of the process.

Gupta Shalvajiran and Nayak's experiments also showed that c-type cytochromes are essential for the growth and fitness of methane-producing Archaea like *M. acetivorans.* The role of c-type cytochromes in methane production varies in different species of Archaea depending on their growth substrate or where they live. These results provide vital information about how Archaea produce methane, and the tools and techniques developed will aid further investigation of the role of Archaea in climate change.

unfolded apo-cyt *c* protein (*Kranz et al., 2009*; *Sanders et al., 2010*). Heme attachment occurs on the extracellular face of the cytoplasmic membrane, and, thus also requires the translocation of the apoprotein and heme across the membrane. The apoprotein is translocated via the Sec pathway (*Sambongi et al., 1996*; *Thöny-Meyer and Künzler, 1997*; *Helde et al., 1997*) and a set of proteins collectively known as the cyt *c* maturation system are involved in energy-dependent heme transport, apoprotein handling, and covalent cofactor attachment (*Kranz et al., 2009*; *Sanders et al., 2010*; *Kranz et al., 1998*). Three distinct systems for cyt *c* maturation have been identified thus far, and have been studied extensively in bacterial and eukaryotic model systems (*Kranz et al., 2009*; *Sanders et al., 2010*). However, despite growing evidence that cyt *c* play a vital role in electron transport processes across members of the Archaea – particularly in microorganisms that metabolize methane and other alkanes – the biogenesis of cyt *c* has never been functionally characterized in an archaeon.

Based on genomic surveys (*Allen et al., 2006*; *Kletzin et al., 2015*; *Chadwick et al., 2021*) of sequenced isolates and metagenomics assembled genomes, the vast majority of cyt *c* containing Archaea encode homologs of the System I cyt *c* maturation pathway (Ccm pathway) that is widespread in members of the Bacteria and is also present in the mitochondria of some Plants and Protozoa (*Kranz et al., 2009*; *Giegé et al., 2008*). The Ccm pathway is best characterized in Gram-negative bacteria, such as *Escherichia coli,* where nine to ten proteins, encoded by *ccmABCDEFGH*(*I*) and *ccdA* (or *dsbD*), are involved in cyt *c* biogenesis (*Figure 1—figure supplement 1*; *Kranz et al., 2009*; *Sanders et al., 2010*). Typically, the *ccm* genes are encoded in an operon on the chromosome and the *ccmH* open reading frame in *E. coli* is often found as two different genes, annotated as *ccmH* (or *ccl2*) and *ccmI* (or *cycH*), in bacteria like *Rhodobacter capsulatus* (*Kranz et al., 2009*). Genetic and biochemical studies of the Ccm pathway have revealed that the membrane-associated CcmABCD complex translocates heme across the cytoplasmic membrane to form holo-CcmE (*Figure 1—figure supplement 1*). Heme from holo-CcmE is ultimately transferred to the apo-cyt *c* by the CcmF/H complex (cytochrome synthetase) to form holo-cyt *c*. CcmG and CcdA (or DsbD) reduce the disulfide bond between cysteine residues of heme-binding motifs (CXXCH) in the apo-cyt *c* prior to heme attachment (*Figure 1—figure supplement 1*; *Fabianek et al., 2000*; *Setterdahl et al., 2000*; *Reid*

*et al., 2001*). Curiously, most, if not all, archaeal genomes sequenced thus far lack several *ccm* genes (namely *ccmD, ccmH, ccmI*) that have been shown to be essential for cyt *c* biogenesis in Bacteria like *E. coli, R. capsulatus, Paracoccus denitrificans, Bradyrhizobium japonicum, Shewenella oneidensis,* and *Desulfovibrio desulfuricans* (*Kranz et al., 1998*; *Thöny-Meyer, 1997*; *Goddard et al., 2010*; *Fu et al., 2015*). In bacteria, CcmD facilitates the release of the heme-bound holo-CcmE from the CcmABCD complex (*Richard-Fogal et al., 2008*), while CcmH and CcmI are a part of the protein complex involved in covalent attachment of heme to the apo-cyt *c* (*Figure 1—figure supplement 1*; *Kranz et al., 2009*; *Sanders et al., 2010*). As such, based on evidence from studies with bacteria, the streamlined Ccm machinery found in archaeal genomes, comprised only of CcmABCEFG and CcdA, would be insufficient for the biogenesis of cyt *c*. However, mature cyt *c* proteins have been identified in many archaeal strains, such as *Haloferax volcanii, Ignicoccus hospitalis, Pyrobaculum islandicum, Ferroglobus placidus,* and *Methanosarcina* spp. (*Kletzin et al., 2015*; *Sreeramulu, 2003*; *Nab et al., 2014*; *Smith et al., 2015*; *Feinberg et al., 2008*; *Wang et al., 2011*). Whether Archaea have replaced CcmD, CcmH, and CcmI with non-orthologous proteins or have reconfigured the Ccm machinery such that *ccmDHI* are no longer essential remains unclear.

In this study, we used the genetically tractable methanogenic archaeon, *M. acetivorans,* as a model system to functionally characterize the pathway for biogenesis of cyt *c* in archaea. Most archaea that encode cyt *c* proteins are either recalcitrant to laboratory cultivation techniques, genetically intractable, or only encode only one cyt *c* that might be essential for growth (*Allen et al., 2006*; *Kletzin et al., 2015*; *Chadwick et al., 2021*; *Arshad et al., 2015*). In contrast, *M. acetivorans* encodes multiple cyt *c* proteins (*Holmes et al., 2019*), can be easily cultivated in a laboratory (*Sowers et al., 1993*), and has state-of-the-art genetic tools (*Guss et al., 2008*; *Kohler and Metcalf, 2012*), including inducible gene expression, tests for gene essentiality, and CRISPR-Cas9-based genome editing for complex genetic manipulation (*Nayak and Metcalf, 2017*). *M. acetivorans* is a methanogen that can only grow by coupling energy conservation to the production of methane gas using acetate or methylated compounds, like methanol, as a growth substrate. At least five different cyt *c* proteins are encoded in the *M. acetivorans* genome and these proteins contain between one and seven heme-binding motifs (*Holmes et al., 2019*). Recent studies have even proposed distinct roles for some of these cyt *c* proteins in intracellular electron transport during methanogenesis (*Wang et al., 2011*; *Li et al., 2006*; *Schlegel et al., 2012*; *Ferry, 2020*), extracellular electron transfer (*Holmes et al., 2019*), as well as direct interspecies electron transfer between *M. acetivorans* and bacterial strains like *Geobacter metallireducens* (*Holmes et al., 2021*). Thus, *M. acetivorans* is not only an ideal candidate to dissect the biogenesis of cyt *c* in Archaea but also to functionally characterize the role of cyt *c* in various electron transfer processes that have ramifications on global carbon cycling as well as on the formation of microbial communities in anoxic environments.

Here, we have used a combination of genetic, molecular, and biochemical analyses, to show that *M. acetivorans* and other archaea use a streamlined version of the Ccm machinery that only requires *ccmABCEF* for cyt *c* biogenesis. To this end, we have shown that the *ccmABCEF* from *M. acetivorans* is sufficient to produce holo-cyt *c* in a heterologous host, *Methanosarcina barkeri* Fusaro, that otherwise is incapable of cyt *c* biogenesis. Our physiological analyses also reveal substrate-specific phenotypes for the cyt *c* biogenesis pathway and cyt(s) *c* during growth and methanogenesis in *M. acetivorans*. A closer inspection of the distribution and synteny of the cyt *c* biogenesis genes in methane-metabolizing archaea related to *Methanosarcina* (i.e. belonging to the order *Methanosarcinales*) suggests that the cyt *c* biogenesis genes were likely acquired in the last common ancestor of the *Methanosarcinales* and have been lost in many extant clades that also do not encode any cyt *c* genes. Although the streamlined Ccm machinery is conserved across Archaea, our evolutionary analyses suggest that the acquisition of cyt *c* biogenesis in Archaea has occurred through multiple horizontal gene transfer (HGT) events with different members of the Bacteria. Overall, we have used the model methanogenic archaeon, *M. acetivorans,* as a model system to characterize a streamlined form of the Ccm machinery used for the biogenesis of cyt *c* in members of Archaea.

## Results

### The Ccm machinery is essential for cyt *c* biogenesis in *M. acetivorans*

We used a sequence-based approach to identify homologs of eight different genes of the Ccm machinery for cyt *c* maturation at four different chromosomal loci in *M. acetivorans*. The *ccmABC* genes (MA1428-MA1430) are located at one chromosomal locus and the *ccmE* gene (MA4149) is in a putative operon with a geranyl farensyl diphosphate synthase at another locus (*Figure 1—figure supplement 1*). Two *ccmF* genes (MA3305 and MA3304 that have been renamed *ccmF1* and *ccmF2*, respectively), which likely represent the C- and N-terminal segments of the bacterial *ccmF* locus, respectively, are encoded in a putative operon. Finally, *ccmG* and *ccdA* are found adjacent to each other on the chromosome (MA4254 and MA4255) (*Figure 1—figure supplement 1*). We used our recently developed Cas9-based genome editing technology to generate markerless in-frame deletion mutants of the *ccmABC* locus, *ccmE*, *ccmF1*, *ccmF2*, *ccmG*, *ccdA*, as well as the Δ*ccmF1*Δ*ccmF2* and Δ*ccmG*Δ*ccdA* double mutants. We picked the Δ*ccmABC* mutant for whole-genome sequencing to screen for off-target CRISPR-Cas9 activity. Apart from the deleted locus, we did not observe any other mutations in the Δ*ccmABC* mutant relative to the parent strain (WWM60) (*Supplementary file 1*). Consistent with our previous observations, these data suggest that off-target activity is negligible during Cas9-mediated genome editing in *M. acetivorans* (*Nayak and Metcalf, 2017*; *Nayak et al., 2017*; *Nayak et al., 2020*).

We selected a representative cyt *c* encoded by the *mmcA* gene (locus tag: MA0658) in the *M. acetivorans* genome to assay for cyt *c* maturation in the *ccm* mutants. MmcA is a heptaheme cyt *c* that contains an N-terminal signal peptide and is likely located in the pseudo-periplasm (*Figure 1—figure*

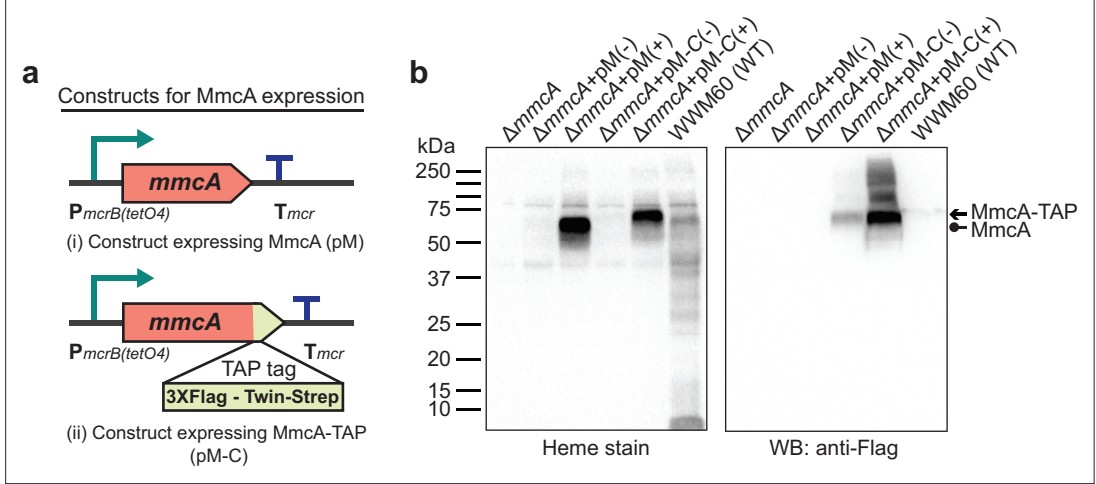

**Figure 1.** Experimental assays to measure the production and maturation of the diagnostic c-type Cytochrome, MmcA. (**a**) The design of diagnostic constructs to use the native heptaheme *c*-type cytochrome, MmcA, to map the cytochrome *c* biogenesis pathway in *Methanosarcina acetivorans*. (i) The control construct (pM) contains an *mmcA* coding sequence whereas (ii) the test construct (pM-C) contains an *mmcA* coding sequence with a C-terminal translational fusion of a tandem affinity purification (TAP) tag comprised of a 3× FLAG sequence and a Twin-Strep sequence. In each construct, gene expression is driven by a tetracycline-inducible medium-strength promoter, P*mcrB*(*tetO4*), and a transcriptional terminator of the *mcr* operon from *M. acetivorans* is provided after the last coding sequence. (**b**) Assays to measure holo-MmcA formation by covalent heme attachment and detect the tagged MmcA protein in whole cell lysates of *M. acetivorans* mutants. (Left) A heme peroxidase-based assay is used to detect the presence of proteins with covalently bound heme in whole cells lysates. Using this assay, both untagged (lane 3) and tagged (lane 5) heme-bound holo-MmcA can be detected upon induction of the genes from each plasmid construct described above. The tagged MmcA protein runs at a higher molecular weight due to the presence of a ca. 7.3 kDa TAP tag at the C-terminus compared to the native MmcA protein. (Right) Immunoblotting with commercial anti-FLAG antibody can be used detect MmcA production in the test construct containing a C-terminal TAP tag fused to the *mmcA* gene (lanes 4 and 5). (-) indicates that no tetracycline was added to the growth medium and (+) indicates that 100 μg/mL tetracycline was added to the growth medium. An equal amount of whole cell lysate protein (80 μg) was loaded in each lane.

The online version of this article includes the following source data and figure supplement(s) for figure 1:

**Source data 1.** Raw gel and blot images.

**Figure supplement 1.** The Ccm Machinery in Bacteria and Archaea.

**Figure supplement 2.** MmcA is a heptaheme c-type cytochrome associated with the Rnf complex in *Methanosarcina acetivorans*.

*supplement 2*). MmcA is a highly expressed cyt *c* that is widely distributed in cyt *c* encoding members of the *Methanosarcinales* (*Holmes et al., 2019*; *Zhou et al., 2021*; *Borrel et al., 2019*) and holo-MmcA is easy to detect as it contains seven heme-binding motifs. Taken together, these functional and technical features make MmcA an ideal candidate to assay cyt *c* biogenesis. To develop a rapid assay for quantification of holo-cyt *c* in the *ccm* mutants, we built a plasmid-based overexpression system by adding a C-terminal TAP (tandem affinity purification) tag comprised of a 3× FLAG sequence and a twin-Strep sequence to the *mmcA* coding sequence placed under the control of a tetracycline-inducible promoter (*Figure 1a*). As a control, we transformed the Δ*mmcA* mutant with an untagged or TAP-tagged cyt *c* overexpression vector to test if the presence of the tag in the MmcA CDS interferes with protein production or heme attachment. To identify protein production, we performed an immunoblot with a commercial anti-Flag antibody. Using this technique, we were able to successfully detect the TAP-tagged MmcA in the cell lysate (*Figure 1b*). To assay for heme attachment (formation of holo-cyt *c*), we used a peroxidase-based heme stain, which detects covalently bound (*c*-type) heme in cyt *c* (*Feissner et al., 2003*). Using this technique, we were able to verify that the TAP-tagged MmcA

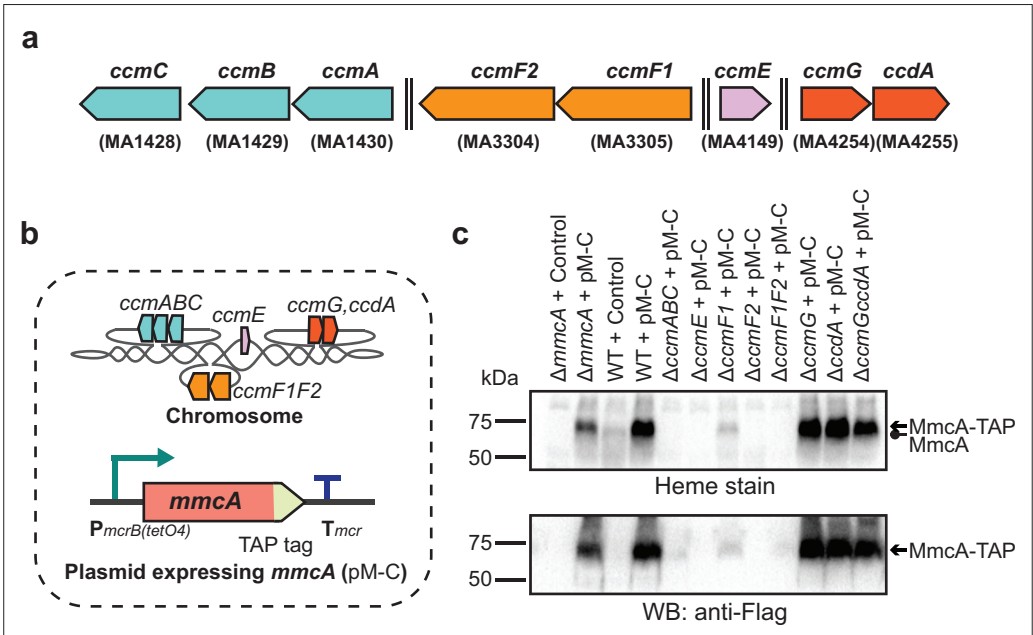

**Figure 2.** Identifying the role of Ccm genes in c-type cytochrome maturation in *Methanosarcina acetivorans*. (**a**) Chromosomal organization of the System I cytochrome *c* maturation machinery (Ccm) genes in *Methanosarcina acetivorans*. Double dashed lines indicate that the genes are located more than 110 kbp away on the chromosome. Genes located at the same chromosomal locus are shown in the same color. (**b**) A schematic showing the genotype of strains used to assay the role of individual *ccm* genes in the maturation of *c*-type cytochromes. In addition to the corresponding *ccm*-deletion on the chromosome, these strains also contain a self-replicating plasmid (pM-C) where the *mmcA* coding sequence with a C-terminal tandem affinity purification (TAP) tag comprised of a 3× FLAG sequence and a Twin-Strep sequence is placed under the control of a tetracycline-inducible medium-strength promoter, P*mcrB*(*tetO4*). In these strains, the expression of the C-terminal TAP-tagged MmcA can be induced by the addition of tetracycline to the growth medium. (**c**) A heme peroxidase-based assay to measure the formation of heme-bound holo-MmcA (top) and Western blots (WB) with anti-FLAG antibody to detect the production of C-terminal TAP-tagged MmcA (bottom). Neither holo-MmcA nor tagged protein could be detected in the Δ*ccmCBA*, Δ*ccmE*, Δ*ccmF2* single mutants, and the Δ*ccmF1*Δ*ccmF2* double mutant suggesting that the apo-MmcA undergoes rapid proteolysis in the absence of a functional Ccm machinery. All assays were conducted with whole cell lysates of cultures grown in medium containing 100 µg/mL tetracycline to induce the expression of C-terminal TAP-tagged MmcA. An equal amount of whole cell lysate protein (60 µg) was loaded in each lane. The vector control (i.e. empty vector) used for this experiment is pJK029A as described previously in *Guss et al., 2008*.

The online version of this article includes the following source data and figure supplement(s) for figure 2:

**Source data 1.** Raw gel and blot images.

**Figure supplement 1.** Alignment of the CcmF amino acid sequence from *Escherichia coli* with the concatenated CcmF2-CcmF1 amino acid sequences from *Methanosarcina acetivorans*.

is capable of undergoing covalent modification to generate the corresponding holo-cyt *c* (*Figure 1b*). Taken together, these controls demonstrate that our overexpression vectors can be used to reliably and rapidly assay the production and covalent modification of MmcA in *M. acetivorans*.

Next, we introduced the plasmid overexpression system with a TAP-tagged MmcA in each of the *ccm*-deletion mutants and assayed for the production of tagged MmcA protein and holo-MmcA in each of the resulting strains (*Figure 2a–c*). We were unable to detect the tagged protein or holo-MmcA in the Δ*ccmABC*, Δ*ccmE*, Δ*ccmF2*, Δ*ccmF1ccmF2* mutants, which suggests that these genes are essential for cyt *c* biogenesis in *M. acetivorans* (*Figure 2c*). Consistent with previous studies, these results also indicate that the apo-MmcA is immediately targeted for degradation in the absence of a functional Ccm machinery (*Goldman et al., 1997*; *Feissner et al., 2006b*; *Allen et al., 2005*). Curiously, we were able to detect faint band corresponding to holo-MmcA in the Δ*ccmF1* mutant (*Figure 2c*). These data suggest that the gene product of *ccmF1* is important but not vital for cyt *c* biogenesis possibly because the catalytically important residues from CcmF in *E. coli* are all present in CcmF2 from *M. acetivorans* (*Figure 2—figure supplement 1*). Finally, we observed that Δ*ccmG* and Δ*ccdA* single mutants as well as the Δ*ccmG*Δ*ccdA* double mutant produced roughly the same amount of holo-MmcA as the parent strain (*Figure 2c*), which indicates that these genes are not essential for disulfide bond reduction in the apo-cyt *c* under highly reducing laboratory growth conditions. Overall, using MmcA as a diagnostic cyt *c*, we have shown that the proteins encoded by *ccmABCEF1F2* in *M. acetivorans* constitute a functional, streamlined version of the Ccm machinery for cyt *c* maturation.

## Archaeal CcmABC is involved in the formation of a holo-CcmE heme chaperone

In the first steps of cyt *c* biogenesis, heme *b* is transported across the cytoplasmic membrane to generate holo-CcmE: a heme chaperone that covalently binds heme and transports it to the cytochrome *c* synthetase complex (*Christensen et al., 2007*; *Feissner et al., 2006a*). In Gram-negative bacteria, like *E. coli*, heme transport and holo-CcmE formation is mediated by the CcmABCD complex (*Figure 3a*). In the CcmABCD complex, CcmD plays an essential role in facilitating the release of holo-CcmE from the CcmABCE adduct (*Kranz et al., 2009*; *Richard-Fogal et al., 2008*). Since *M. acetivorans* and all other sequenced archaeal strains lack CcmD, we investigated the role of the CcmABC complex in the formation of holo-CcmE (*Figure 3b*). To this end, we developed a plasmid-based overexpression system for CcmE with a C-terminal 1× Strep-1× Flag tag placed under the control of a tetracycline-inducible promoter (*Figure 3c*). We introduced the C-tagged CcmE over-expression plasmid in the parent strain (WWM60) as well as the Δ*ccmABC* and Δ*ccmE* mutants and were able to detect tagged-CcmE in protein enriched from the membrane fraction (using an anti-Strep-affinity column) by immunoblotting with an anti-FLAG antibody in all plasmid complemented strains (*Figure 3d*). We were only able to observe the heme-bound holo-CcmE by heme staining the enriched membrane fraction of the plasmid complemented parent strain (WWM60) and Δ*ccmE* mutant (*Figure 3d*). Furthermore, we were unable to detect holo-cyt *c*, including the MmcA protein, by heme staining the total cell lysate of the Δ*ccmABC* strain complemented with the CcmE overexpression plasmid, which indicates that overexpression of CcmE does not rescue the *ccmABC* lesion in the cytochrome maturation machinery (*Figure 3—figure supplement 1*). These data support the hypothesis that the CcmABC complex transports heme to form holo-CcmE in *M. acetivorans*.

## A CXXXY motif in archaeal CcmE is required for heme attachment and protein stability

In bacteria with a Ccm machinery, like *E. coli,* a conserved histidine residue (H130) present in the HXXXY motif of CcmE covalently binds heme (*Schulz et al., 1998*). Notably, all sequenced archaea (and a few bacteria like *D. desulfuricans*) have replaced this histidine with a cysteine (*Figure 4a*; *Allen et al., 2006*; *Goddard et al., 2010*). To test if the cysteine residue (C120) in the conserved motif (CXXXY) of CcmE in *M. acetivorans* has a functional role similar to the histidine residue in *E. coli* or the cysteine residue in *D. desulfuricans*, we built a plasmid-based overexpression system for mutant alleles of *ccmE* with either a C120A or a C120H substitution and a C-terminal 1× Strep-1× Flag tag under the control of an inducible promoter (*Figure 4a and b*). We introduced these plasmids in the Δ*ccmE* mutant and successfully detected the wild-type and C120A CcmE protein in the Strep-enriched membrane fraction by immunoblotting with anti-FLAG antibodies (*Figure 4c*). Despite repeated attempts, we

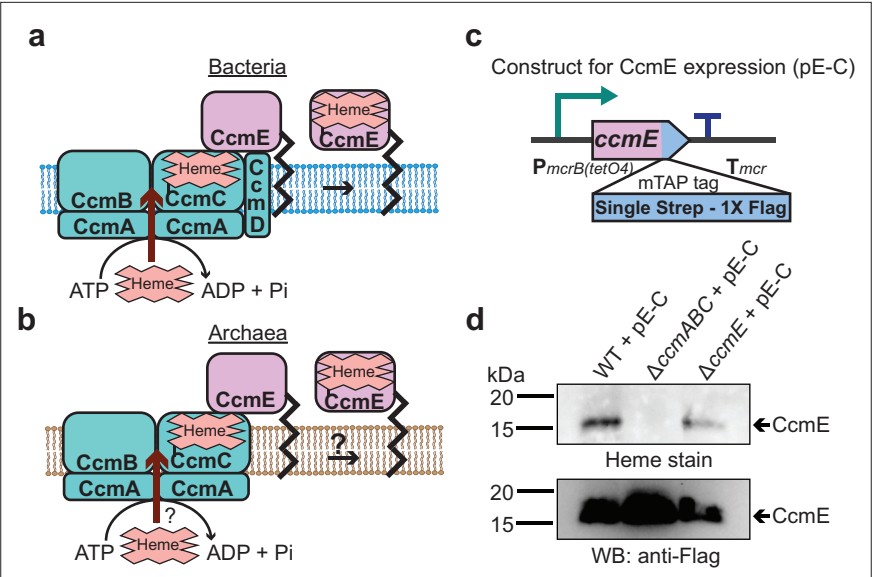

**Figure 3.** CcmABC is required for the heme transport to form holo-CcmE in *Methanosarcina acetivorans*. (**a**) The formation of heme-bound holo-CcmE in bacteria containing the System I Ccm machinery is facilitated by the CcmABCD complex. The ATP-dependent CcmABC complex translocates heme *b* from the cytosol to CcmE and CcmD facilitates the dissociation of holo-CcmE from the CcmABCDE adduct. (**b**) Since archaea lack the *ccmD* gene, the formation of holo-CcmE is unclear. One hypothesis, shown here, is that the CcmABC complex in archaea mediates the dissociation of holo-ccmE independent of CcmD. (**c**) The design of a test construct (pE-C) containing the *ccmE* coding sequence with C-terminal translational fusion of a modified tandem affinity purification (mTAP) tag comprising of a Single-Strep sequence and a 1× FLAG sequence. In this construct, the coding sequence is placed in-between a tetracycline-inducible medium-strength promoter, P*mcrB*(*tetO4*) (**Guss et al., 2008**), and the transcriptional terminator of the *mcr* operon from *Methanosarcina acetivorans*. Here, the expression of the C-terminal mTAP-tagged CcmE can be induced by the addition of tetracycline to the growth medium. (**d**) A heme peroxidase-based assay to measure the formation of holo-CcmE (top) and a Western blot (WB) with anti-FLAG antibody to detect the production of C-terminal mTAP-tagged CcmE (bottom) was conducted. Even though CcmE is produced in the Δ*ccmABC* mutant, holo-CcmE cannot be formed, consistent with the model shown in (**b**). All assays were performed with an enriched membrane fraction (i.e. protein eluted after passing the membrane fraction through a Strep-affinity column) of cultures grown in medium containing 100 µg/mL tetracycline to induce the expression of C-terminal mTAP-tagged CcmE. An equal amount of protein (1 µg) was loaded in each lane.

The online version of this article includes the following source data and figure supplement(s) for figure 3:

**Source data 1.** Raw gel and blot images.

**Figure supplement 1.** Expression of CcmE with a C-terminal 1× FLAG-1× Strep tag from a plasmid (pE) in the parent strain of *Methanosarcina acetivorans* (WWM60) and the Δ*ccmE* background leads to the production of heme-bound holo-MmcA as detected by a heme peroxidase assay of whole cell lysates (lanes 1, 3 of heme-stained gel).

**Figure supplement 1—source data 1.** Raw gel and blot images.

were unable to detect the C120H CcmE mutant in the membrane or soluble fraction (*Figure 4c* and *Figure 4—figure supplement 1*). This outcome is consistent with the hypothesis that the C120H substitution considerably destabilizes CcmE, leading to proteolysis of the gene product. Next, we were able to observe the wild-type CcmE by heme staining the Strep-enriched membrane fraction of the corresponding strain however, no heme-stained band corresponding to CcmE was detected for the C120A mutant (*Figure 4c*). Furthermore, we were unable to detect holo-MmcA or any other holo-cyt *c* by heme staining the total cell lysate of the Δ*ccmE* strain complemented with either the C120A or C120H CcmE (*Figure 4—figure supplement 2*). These results provide strong evidence in support of the hypothesis that the C120 residue of the CXXXY motif is important for heme attachment as well as protein stability in the archaeal CcmE.

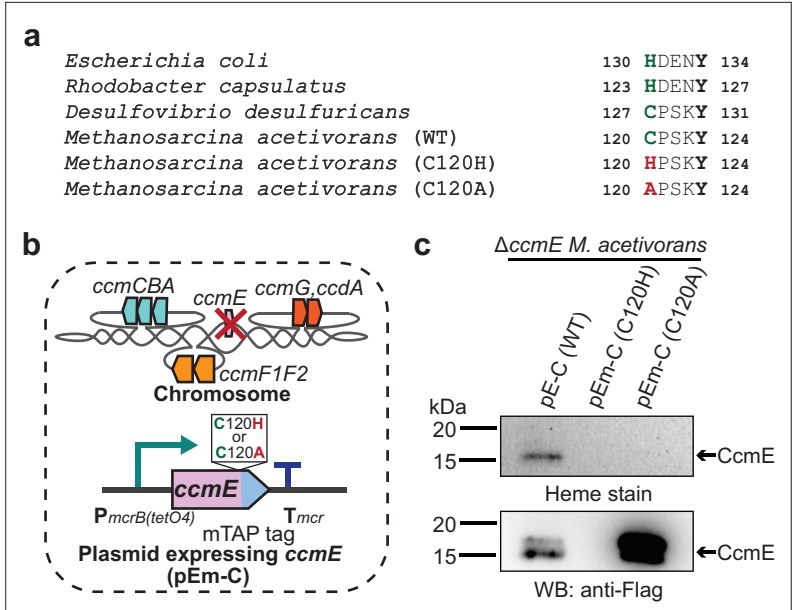

**Figure 4.** Identifying the heme-binding residue in CcmE from *Methanosarcina acetivorans*. (**a**) Alignment of the heme-binding domain in CcmE sequences derived from representative bacteria and archaea. The heme-binding residue in bacteria can vary: some species like *Escherichia coli* covalently bind heme using a histidine residue whereas others like *Desulfovibrio desulfuricans* covalently bind heme using a cysteine residue. Almost all archaeal CcmE sequences, including the sequence derived from *Methanosarcina acetivorans,* contain a conserved cysteine residue that is likely involved in heme binding. (**b**) To test the role of the cysteine residue in the CcmE sequence derived from *M. acetivorans,* we expressed the C120A and C120H point mutants of *ccmE* with a modified tandem affinity purification (mTAP) tag comprising of a Single-Strep sequence and a 1× FLAG sequence at the C-terminus as described in *Figure 3c*. (**c**) A heme peroxidase-based assay was used to measure the formation of holo-CcmE (top) and Western blot (WB) with anti-FLAG antibody was used to detect the production of C-terminal mTAP-tagged CcmE (bottom). Neither the C120A or the C120H mutants of CcmE contain covalently bound heme (top); furthermore, no mTAP-tagged CcmE was detected by WB for the C120H mutant (bottom), possibly indicating that this point mutation destabilizes the protein substantially. All assays were performed with an enriched membrane fraction (i.e. protein eluted after passing the membrane fraction through a Strep-affinity column) of cultures grown in medium containing 100 µg/mL tetracycline to induce the expression of C-terminal mTAP-tagged CcmE. An equal amount of protein (1 µg) was loaded in each lane.

The online version of this article includes the following source data and figure supplement(s) for figure 4:

**Source data 1.** Raw gel and blot images.

**Figure supplement 1.** Expression of the C120H mutant of CcmE from *Methanosarcina acetivorans* with a C-terminal 1× FLAG-1× Strep tag in the Δ*ccmE* background does not lead to the production of any CcmE protein that can be detected by heme peroxidase assays (top gel) or Western blot using an anti-Flag antibody (bottom gel) in the Strep-enriched membrane fraction (lane 1) or the soluble fraction (lane 2).

**Figure supplement 1—source data 1.** Raw gel and blot images.

**Figure supplement 2.** Expression of the wild-type (WT) sequence of CcmE from *Methanosarcina acetivorans* with a C-terminal 1× FLAG-1× Strep tag (C-tagged CcmE) in the Δ*ccmE* background leads to the production of heme-bound holo-MmcA as detected by a heme peroxidase assay of whole cell lysates (lane 1 of heme-stained gel).

**Figure supplement 2—source data 1.** Raw gel and blot images.

## A streamlined Ccm machinery comprised of CcmABCEF is necessary and sufficient for cyt *c* biogenesis in *Methanosarcina* spp

While our genetic studies with *M. acetivorans* clearly show that the proteins encoded by *ccmABC, ccmE, ccmF1,* and *ccmF2* are essential for cyt *c* biogenesis, they do not inform us of any non-orthologous proteins that might also be involved in this process. To test if the cyt *c* maturation pathway encoded by the *ccm* genes in *M. acetivorans* is both necessary and sufficient for cyt *c* biogenesis, we expressed *mmcA* and a synthetic operon comprised of *ccmABCEF1F2* in a heterologous host. We chose *M.*

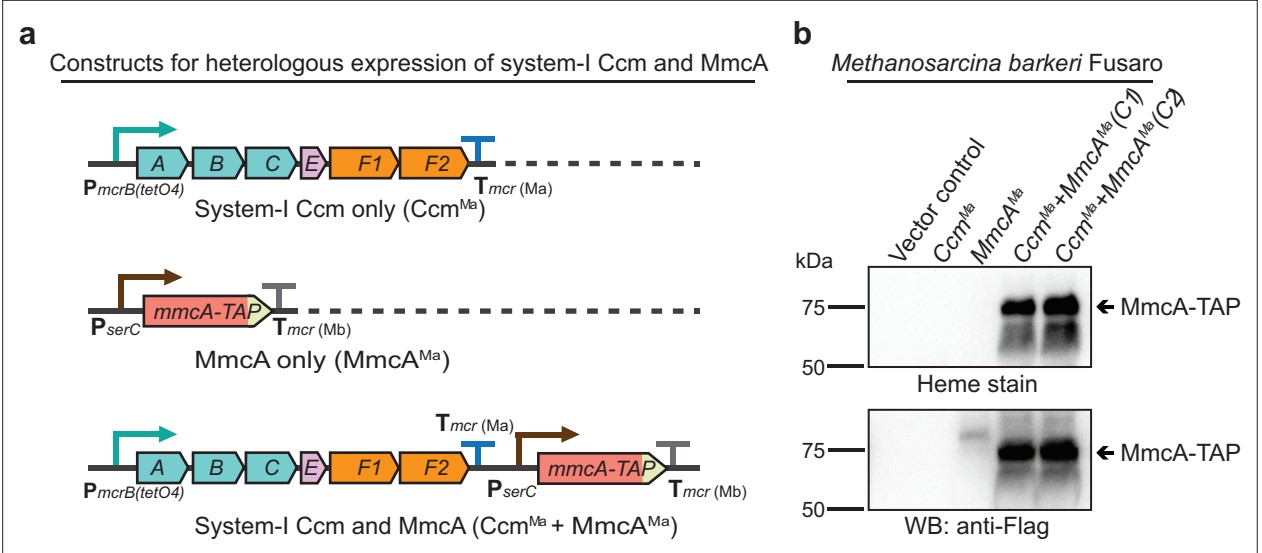

**Figure 5.** Heterologous expression of the Ccm machinery and the diagnostic c-type cytochrome MmcA in *Methanosarcina barkeri* Fusaro. (**a**) The design of constructs to express a synthetic operon comprised of the *ccmABCEF1F2* genes from *Methanosarcina acetivorans* and/or the *mmcA* coding sequence from *M. acetivorans* with a C-terminal tandem affinity purification (TAP) tag comprised of a 3× FLAG sequence and a Twin-Strep sequence. In these constructs, the expression of the *ccmABCEF1F2* operon is driven by a tetracycline-inducible medium-strength promoter, P*mcrB(tetO4)*, and a transcriptional terminator of the *mcr* operon from *M. acetivorans* T*mcr*(Ma) is provided after the last coding sequence. Each gene, apart from *ccmA*, contains its native Shine-Dalgarno sequence. The C-terminal TAP-tagged MmcA is placed under the control of a constitutive medium-strength promoter (P*serC*) and the transcriptional terminator of the *mcr* operon from *Methanosarcina barkeri* T*mcr*(Mb) has been added immediately downstream of the coding sequence. These constructs are integrated at the ØC31 attachment site at a neutral locus on the *M. barkeri* Fusaro. (**b**) Heme peroxidase assays to detect the production of heme-bound Holo-MmcA (top) and Western blots (WB) to detect C-terminal TAP-Tagged MmcA protein (bottom) in whole cell lysates of *M. barkeri* Fusaro cultures grown in medium containing 100 µg/mL tetracycline. A diagnostic *c*-type cytochrome like MmcA can be produced in a heterologous host lacking a native *c*-type cytochrome maturation machinery (*M. barkeri*) by expressing the *ccmABCEF1F2* genes from *M. acetivorans* (lanes 4–5). $C_1$ and $C_2$ refer to two independently colonies of the *M. barkeri* Fusaro cointegrate expressing the Ccm machinery and MmcA from *M. acetivorans*. An equal amount of whole cell lysate (60 µg) was loaded in each lane. The vector control (i.e. empty vector) used for this experiment is pJK029A as described elsewhere (*Guss et al., 2008*).

The online version of this article includes the following source data and figure supplement(s) for figure 5:

**Source data 1.** Raw gel and blot images.

**Figure supplement 1.** Measuring c-type cytochrome production in cell lysates from *Methanosarcina acetivorans* and *Methanosarcina barkeri*.

**Figure supplement 1—source data 1.** Raw gel and blot images.

barkeri Fusaro as a heterologous host for this study as it has the same codon usage pattern as *M. acetivorans*, contains an intact pathway for heme *b* synthesis, and is known to produce *b* type cytochromes, but does not encode any of the *ccm* genes or cyt *c* in its genome. To confirm that *M. barkeri* Fusaro does not produce cyt *c*, we assayed the cell lysate with the peroxidase-based heme stain for cyt *c* described earlier. Using this technique, we did not detect any signal of cyt *c* in *M. barkeri* Fusaro (*Figure 5—figure supplement 1*). We built a plasmid-based expression system containing some or all of the following components: (i) *ccmABCEF1F2* genes under the control of a tetracycline-inducible promoter and (ii) *mmcA* (a diagnostic cyt *c*) with a C-terminal TAP tag under the control of a constitutive promoter (*Figure 5a*). Each of these plasmids was integrated on the *M. barkeri* Fusaro chromosome at a neutral locus using a ØC31 integrase system described previously (*Guss et al., 2008*). We were able to detect a band corresponding to the TAP-tagged MmcA by immunoblotting with anti-FLAG antibodies and heme staining in *M. barkeri* strains expressing both *mmcA* and *ccm* genes (*Figure 5b*). These results clearly indicate that the *ccmABCEF1F2* genes in *M. acetivorans* encode a complete, functional, and streamlined version of the System I cytochrome *c* maturation machinery previously characterized in Bacteria.

## The production of cyt(s) *c* is important for growth of *M. acetivorans*

Multiple cyt(s) *c* are encoded in the genome of *M. acetivorans* and some of them, like MmcA, are associated with the electron transport chain; yet the cyt *c* maturation machinery does not seem to be

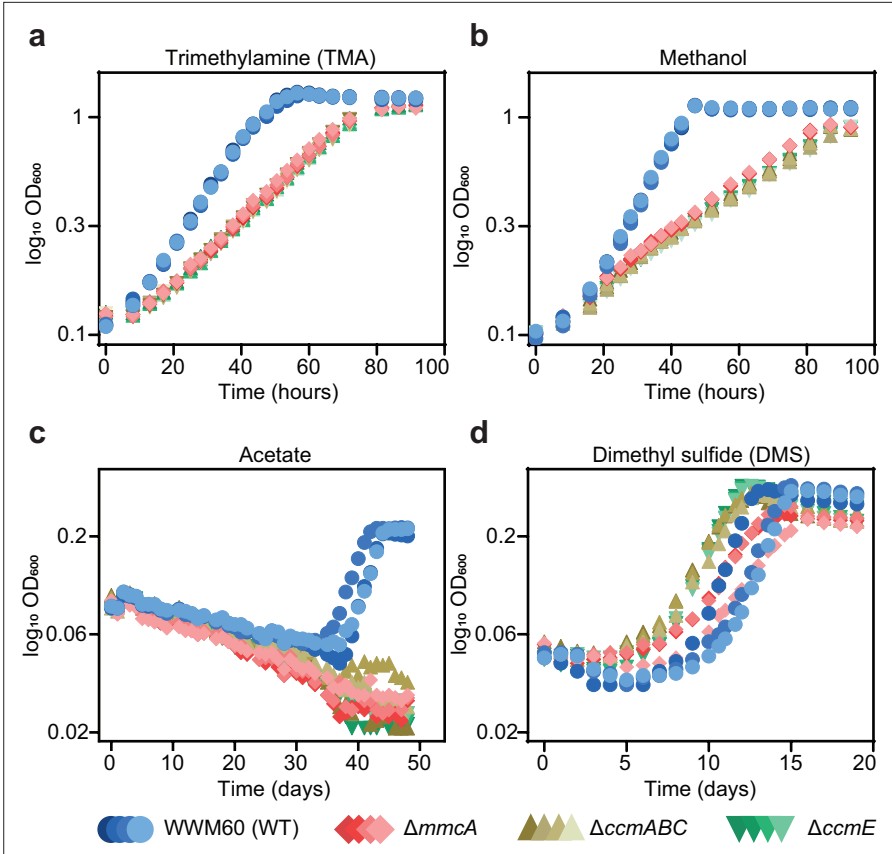

**Figure 6.** Growth curves of the parent strain (WWM60; referred to as wild-type [WT]) (in blue circles), the Δ*mmcA* mutant (in red diamonds), the Δ*ccmABC* mutant (in olive triangles), and the Δ*ccmE* mutant (in green inverted triangles) in high-salt minimal medium with (**a**) 50 mM trimethylamine hydrochloride (TMA), (**b**) 125 mM methanol, (**c**) 40 mM sodium acetate (acetate), and (**d**) 20 mM dimethyl sulfide (DMS) as the sole carbon and energy source. Four replicates were used for growth assays on TMA and methanol and three replicates were used for growth assays on acetate and DMS.

The online version of this article includes the following figure supplement(s) for figure 6:

**Figure supplement 1.** Different modes of methanogenic growth in *Methanosarcina acetivorans*.

essential under standard growth conditions (i.e. in medium with trimethylamine hydrochloride (TMA) as the sole carbon and energy source). To determine the physiological role of the Ccm machinery as well as the cyt(s) *c* produced by this process, we assayed the growth characteristics of the Δ*mmcA*, Δ*ccmE*, and Δ*ccmABC* mutants on growth substrates that represent a variety of methanogenesis pathways and thermodynamic regimes for *M. acetivorans* (**Figure 6** and **Figure 6—figure supplement 1**). In these analyses, the Δ*ccmE* and Δ*ccmABC* mutants represent lesions in the Ccm pathway that would prevent maturation of all expressed cyt *c,* whereas the Δ*mmcA* mutant represents an in-frame deletion in a specific cyt *c* found in the membrane-associated Rnf (Rhodobacter nitrogen fixation) complex that catalyzes the transfer of electrons between ferredoxin and methanophenazine (a membrane-bound electron carrier) coupled to Na$^+$ translocation (***Wang et al., 2011***; ***Schlegel et al., 2012***; ***Figure 6—figure supplement 1b***). We assayed growth on substrates that represent the two modes of methanogenesis in *M. acetivorans*: the methylotrophic pathway and the aceto-clastic pathway. During methylotrophic growth, on compounds like TMA, methanol or dimethyl sulfide (DMS), cells disproportionate the methylated compound to produce carbon dioxide and methane in a 3:1 ratio (***Figure 6—figure supplement 1a and b*** and **1d**). In contrast, during growth on acetate, via the acetoclastic pathway, the substrate undergoes dismutation to produce carbon dioxide and methane in a 1:1 ratio (***Figure 6—figure supplement 1c***). Furthermore, the Gibbs free energy change (ΔG°′) of these substrates, from –164.5 kJ/mol TMA to –36.0 kJ/mol acetate, also captures the entire bioenergetic landscape of methanogenesis.

We observed substantially slower growth than the parent strain for the *ccm*-deletion mutants (Δ*ccmABC* and Δ*ccmE*) as well as the cyt *c* deletion mutant (Δ*mmcA*) under all conditions tested; however, the magnitude of the growth defect varied dramatically (*Figure 6*; *Supplementary file 2*). In growth medium with acetate as the sole substrate, we did not detect any growth for the *ccm*-deletion mutants or the Δ*mmcA* mutant (over the course of 7 weeks), which indicate that holo-MmcA and, possibly, other cyt *c* produced by the Ccm pathway are essential for acetoclastic methanogenesis (*Figure 6c*; *Supplementary file 2*). In contrast, cyt *c* are not essential but important during growth on methylated compounds, however, the growth characteristics of each mutant varied substantially (*Figure 6a, b and d*; *Supplementary file 2*). Notably, despite a shorter lag, the *ccm*-deletion mutants grew slower than the parent strain on DMS (*Figure 6d*). On TMA and DMS, the *ccm*-deletion mutants as well as the Δ*mmcA* mutant had a slower growth rate compared to the parent (WWM60) but were indistinguishable from each other (*Figure 6a and d*; *Supplementary file 2*). Taken together, these data indicate that although MmcA is dispensable, it is the sole physiologically relevant cyt *c* during growth on TMA and DMS. In contrast, on methanol, the *ccm*-deletion mutants grew significantly slower than the parent (WWM60) compared to the Δ*mmcA* mutant (*Figure 6b*; *Supplementary file 2*). These data suggest that MmcA is the predominant cyt *c* synthesized on methanol, and is vital for growth, but other cyt *c* encoded in the genome also play a minor but physiologically relevant role. Our growth analyses underscore the importance of cyt *c* on the physiology of *M. acetivorans* and highlight that the role of individual cyt(s) *c* can vary in a substrate-specific manner.

## Evolutionary analysis of cyt *c* biogenesis in archaea

Even though cyt *c* are important for optimal growth and methanogenesis in *M. acetivorans,* the Ccm machinery and cyt *c* proteins are absent in the close relative *M. barkeri* Fusaro. This uneven distribution of cyt *c* in strains within the genus *Methanosarcina* could be due to gene loss in strains like *M. barkeri* Fusaro or gene gain by HGT of the Ccm machinery and cyt *c* genes in strains like *M. acetivorans*. To distinguish between the two evolutionary hypotheses, we mapped the distribution of the Ccm machinery on the species tree of strains belonging to the genus *Methanosarcina* that are present in the Genome Taxonomy Database (GTDB) (*Parks et al., 2022*). The species tree is based on core genome alignment of the corresponding strains and was obtained from AnnoTree (*Mendler et al., 2019*; *Figure 7a*). Either all genes of the Ccm machinery are present in a strain or the whole machinery is completely absent (*Figure 7a*). The strains that encode the Ccm machinery are not monophyletic, that is, the Ccm machinery is not just present in one single clade of *Methanosarcina* strains. This observation strongly supports the hypothesis that the Ccm machinery was present in the last common ancestor of extant *Methanosarcina* spp. and strains like *M. barkeri* Fusaro have lost these genes (*Figure 7a*). This hypothesis is further corroborated by the synteny of the chromosomal locus surrounding the *ccmABC, ccmE,* and *ccmF1ccmF2* genes across all *Methanosarcina* spp. (*Figure 7b*). Notably, strains like *M. barkeri* MS and *M. vacuolata* contain a truncated ORF (192 and 195 bp, respectively) that has sequence homology to the C-terminus of *ccmE* likely as a scar of a gene loss event (*Figure 7a and b*). To test whether the Ccm machinery was acquired by HGT in the last common ancestor of extant strains within the genus *Methanosarcina* or is present in other genera within the family *Methanosarcinaceae*, we mapped the distribution of the Ccm machinery (i.e. the presence of the *ccmABCEF1F2* genes) across the *Methanosarcinaceae* (*Figure 7—figure supplement 1*). The Ccm machinery is broadly distributed in methanogens and anaerobic methanotrophic (ANME) archaea across the *Methanosarcinaceae* suggesting an important role for cyt *c* in methane metabolism across this clade (*Figure 7—figure supplement 1*). While the Ccm machinery is universally conserved in some genera, like *Methanococcoides, Methanosalsum,* and *Methanohalobium*, this whole machinery seems to have been lost in methanogenic isolates from members of other genera, like *Methanosarcina, Methanolobus,* and *Methanomethylovorans* (*Figure 7—figure supplement 1*). As ANME archaeal genomes are assembled from metagenomes and often incomplete, it is impossible to infer gene loss events in these organisms. Taken together, while widely distributed, the Ccm machinery and cyt *c* proteins in methanogens within the *Methanosarcinaceae* are lost frequently likely as an adaptation to certain environmental conditions.

To test if the Ccm machinery in members of the *Methanosarcinaceae* was vertically inherited from an archaeal ancestor or acquired through inter-domain HGT from bacterial clades, we constructed maximum-likelihood phylogenetic trees of each *ccm* gene from *M. acetivorans* (*Figure 8* and

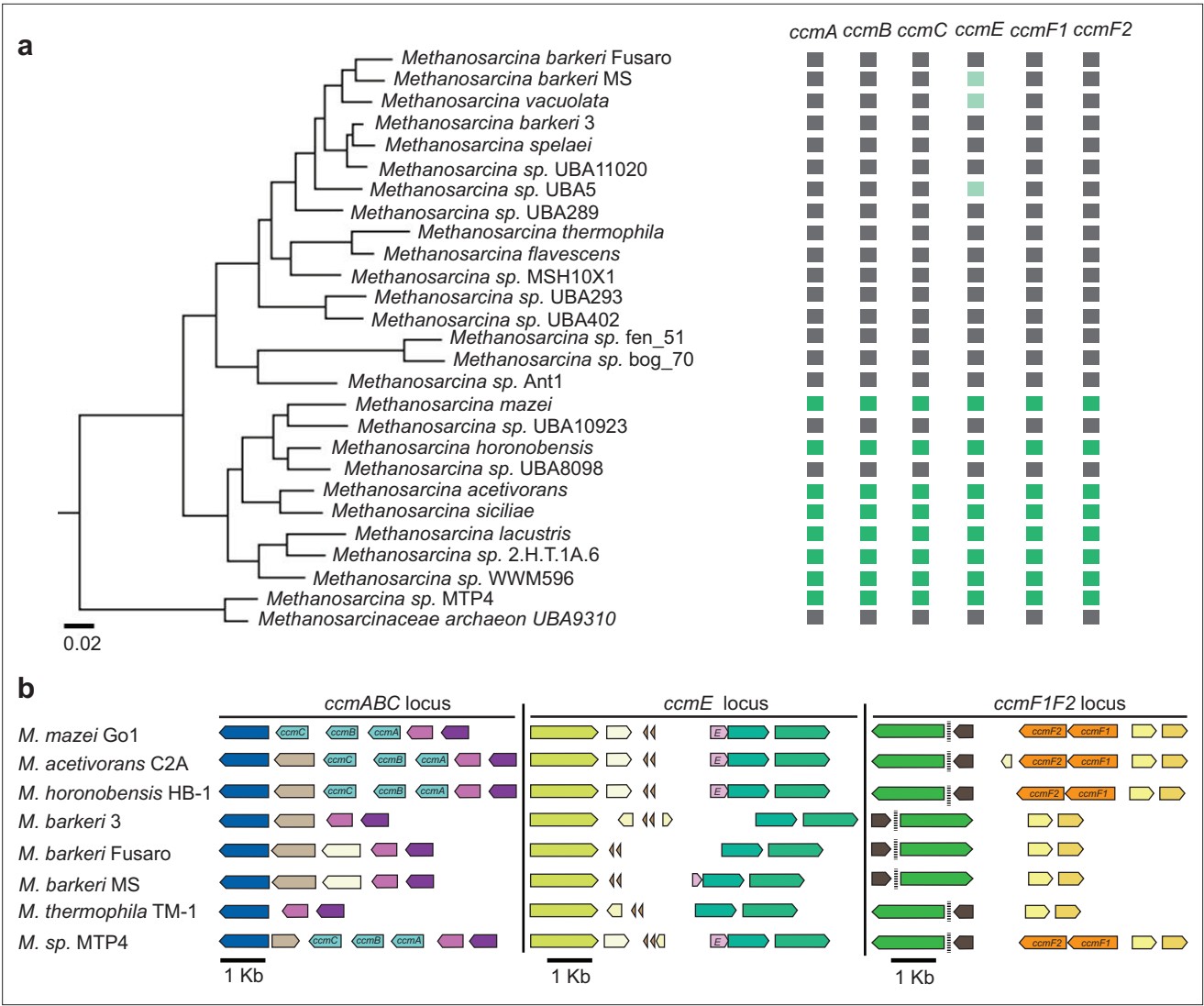

**Figure 7.** Distribution of the Ccm machinery genes in members of the *Methanosarcina* Genus. (**a**) A phylogenetic tree for strains belonging to the genus *Methanosarcina* in the Genome Taxonomy Database (GTDB) was obtained from AnnoTree (**Mendler et al., 2019**). The presence or absence of genes functionally annotated as *ccmA, ccmB, ccmC, ccmE, ccmF1,* and *ccmF2* in the genome of individual strains is shown in dark green (present) or black (absent), respectively. A few strains like *M. barkeri* MS, *M. vacuolata,* and *M.* sp. UBA5 encode a truncated copy of *ccmE* as indicated in light green. (**b**) Chromosomal organization of genes surrounding the *ccmABC, ccmE,* and *ccmF1ccmF2* locus in *Methanosarcina* strains. Genes of the same color (except light yellow) represent members of the same orthologous group. Synteny of the *ccmABC, ccmE,* and *ccmF1ccmF2* chromosomal locus across members of the genus *Methanosarcina* supports the hypothesis that these genes were present in the last common ancestor and have been lost in several extant lineages.

The online version of this article includes the following figure supplement(s) for figure 7:

**Figure supplement 1.** A phylogenomic reconstruction of the relationship between various genera (depicted by different colors along the circumference of the radial tree) within the family *Methanosarcinaceae* from AnnoTree (Mendler et al., 2019).

*Figure 8—figure supplements 1–6*). For each of the *ccm* genes, one or more clades comprising of strains belonging to the order *Methanosarcinales* is nested within clades derived from different bacterial groups (*Figure 8* and *Figure 8—figure supplements 1–6*). Based on these trees, it is evident that Ccm machinery in the *Methanosarcinales* was acquired by HGT from bacteria. It is also likely that different *ccm* genes were acquired from different bacteria, which is further corroborated by the observation that these genes are not present in an operon contrary to their chromosomal arrangement in bacterial strains (*Figure 1—figure supplement 1* and *Figure 8—figure supplements 1–6*). Taken together, the streamlined Ccm machinery, which lacks CcmDHI, is not ancestral to the Archaea, rather it has been acquired through multiple inter-domain HGT events between Archaea and Bacteria. While

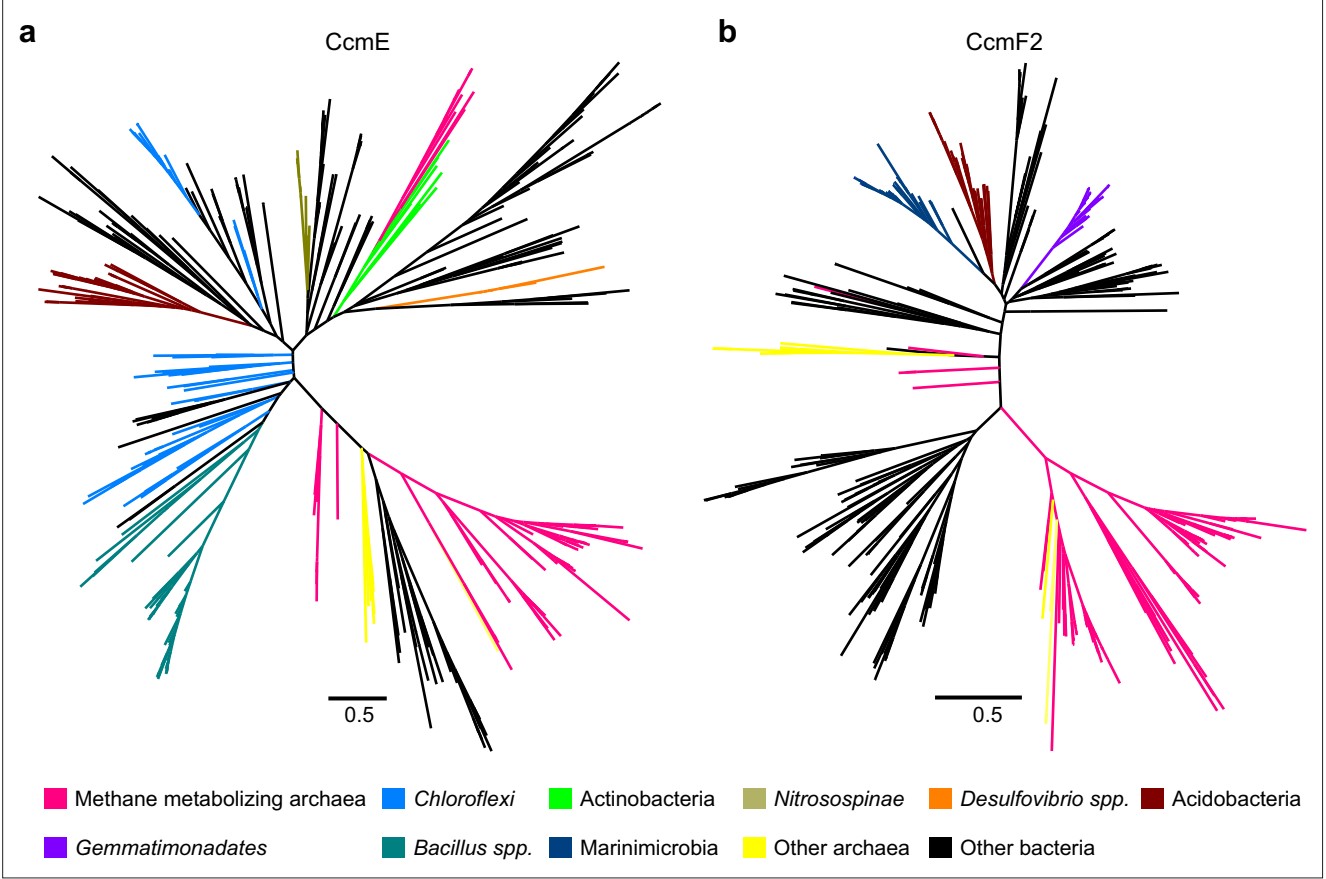

**Figure 8.** Maximum-likelihood phylogenetic trees of 500 (**a**) CcmE and (**b**) CcmF2 sequences obtained from the NCBI non-redundant (nr) protein database using the corresponding sequence from *Methanosarcina acetivorans* as the search query. Sequences belonging to certain functional groups or evolutionarily related groups of archaea and bacteria are shown in colors denoted in the legend at the bottom. These gene trees are indicative of multiple horizontal gene transfer (HGT) events between members of the Archaea and Bacteria.

The online version of this article includes the following figure supplement(s) for figure 8:

**Figure supplement 1.** An unrooted maximum-likelihood tree of 500 CcmA sequences retrieved from the NCBI non-redundant protein sequence database using the corresponding sequence from *Methanosarcina acetivorans* as the search query.

**Figure supplement 2.** An unrooted maximum-likelihood tree of 500 CcmB sequences retrieved from the NCBI non-redundant protein sequence database using the corresponding sequence from *Methanosarcina acetivorans* as the search query.

**Figure supplement 3.** An unrooted maximum-likelihood tree of 500 CcmC sequences retrieved from the NCBI non-redundant protein sequence database using the corresponding sequence from *Methanosarcina acetivorans* as the search query.

**Figure supplement 4.** An unrooted maximum-likelihood tree of 500 CcmE sequences retrieved from the NCBI non-redundant protein sequence database using the corresponding sequence from *Methanosarcina acetivorans* as the search query.

**Figure supplement 5.** An unrooted maximum-likelihood tree of 500 CcmF1 sequences retrieved from the NCBI non-redundant protein sequence database using the corresponding sequence from *Methanosarcina acetivorans* as the search query.

**Figure supplement 6.** An unrooted maximum-likelihood tree of 500 CcmF2 sequences retrieved from the NCBI non-redundant protein sequence database using the corresponding sequence from *Methanosarcina acetivorans* as the search query.

it is not possible to rule out the role of historical contingency in the notable absence of CcmDHI, it is tempting to speculate that unique features of the archaeal cell render CcmHDIG and CcdA dispensable such that a streamlined Ccm machinery comprising of CcmABCEF is sufficient for cyt *c* maturation in Archaea.

## Discussion

Cyt *c* are widely found in members of the Archaea where they catalyze numerous electron transfer processes that facilitate microbial interactions and fuel biogeochemical cycles. The biogenesis of cyt *c* is complex and is mediated by a dedicated pathway that can vary substantially across the tree of life. In the vast majority of eukaryotes, a single protein called cyt *c* heme lyase is involved in the biogenesis of cyt *c* (*Kranz et al., 2009*). In contrast, bacteria contain more complex pathways that are fine-tuned to work with lower concentrations of heme (*Kranz et al., 2009*). The most complex of these pathways, called the Ccm machinery, requires at least nine to ten proteins (CcmABCDEFGH(I) and CcdA/DsbD) that work in concert for the biogenesis of cyt *c*. The biochemical details of the Ccm machinery in bacteria has been studied in great detail using model systems over the past few decades and the role of each *ccm* gene is well established at this point (*Kranz et al., 2009*; *Sanders et al., 2010*). Previous research has shown that archaea encode some components of the bacterial Ccm machinery (*Allen et al., 2006*; *Kletzin et al., 2015*), however, orthologs of at least three proteins (CcmD, CcmH, and CcmI) that are absolutely critical for cyt *c* maturation in bacteria are universally absent in archaeal genomes. Thus, it was unclear how the Ccm machinery is being repurposed (if at all) for cyt *c* biogenesis in archaea. In this study, we use the methanogen *M. acetivorans* to characterize a streamlined form of the Ccm system comprised of only five proteins (CcmABCEF) that likely serves as the predominant route for cyt *c* biogenesis in members of the Archaea (*Figures 2 and 5*).

Based on our findings, there are three key differences between the Ccm machinery in Bacteria and Archaea. While *ccmD* seems to be uniformly absent in archaeal genomes (*Allen et al., 2006*), it is universally present in bacterial genomes and is essential for cyt *c* biogenesis in a number of strains ranging from *E. coli* and *Rhodobacter capsultus* to *D. desulfuricans* (*Kranz et al., 2009*; *Goddard et al., 2010*; *Richard-Fogal et al., 2008*). In bacteria, CcmD is a small (69 aa long) transmembrane protein that associates with CcmABC and facilitates the release of holo-CcmE in the periplasm after heme transport and attachment (*Richard-Fogal et al., 2008*; *Feissner et al., 2006a*). In *M. acetivorans,* the CcmABC complex is capable of forming holo-CcmE and releasing it into the pseudo-periplasm in a CcmD-independent manner (*Figure 3*). It is unlikely that the *ccmD* gene has been fused to *ccmA*, *ccmB*, or *ccmC* based on the sequence length and alignment of the archaeal genes with their bacterial counterparts (*Figure 8—figure supplements 1–3*). It is also unlikely that the CcmD-independent mechanism for the release of holo-CcmE from CcmABC might be linked to the CXXXY heme-binding motif in CcmE because some bacteria, like *D. desulfuricans*, that encode *ccmD* and require this gene product for cyt *c* biogenesis also contain a CXXXY heme-binding motif in CcmE (*Figure 4*; *Goddard et al., 2010*). One plausible hypothesis for the absence of CcmD might be the unusual architecture of archaeal membranes. The archaeal cytoplasmic membrane is composed of isoprenoid lipids (phytanyl units), compared to phospholipids in bacterial cytoplasmic membranes. This lipid composition might alter how CcmABC and CcmE are anchored into the membrane and render CcmD inconsequential during the formation and release of holo-CcmE.

Next, the archaeal CcmE universally contains a CXXXY heme-binding motif whereas bacterial CcmE can either have an HXXXY motif or a CXXXY motif depending on the strain (*Kranz et al., 2009*; *Goddard et al., 2010*). In *E. coli,* an H130C allele of CcmE can still bind heme but cannot transfer the heme to the CcmFH complex (*Goddard et al., 2010*; *Enggist and Thöny-Meyer, 2003*; *Stevens et al., 2003*). In contrast, a C120H mutation in *M. acetivorans* seems to destabilize the protein to such an extent that it can no longer be detected (*Figure 4* and *Figure 4—figure supplement 1*). Similarly, no heme-bound CcmE was detected for the C127H allele of CcmE derived from the bacterium *D. desulfuricans* (*Goddard et al., 2010*). Taken together, these data suggest that the Cys and His residues involved in heme-binding CcmE are not interchangeable. In fact, many other as yet uncharacterized features of CcmE, possibly unique to these two groups of CcmE proteins, are likely to be equally important for the formation of holo-CcmE and the transfer of heme to the apo-cyt *c* via interactions between CcmE and CcmF as well as other proteins. Whether the absence of an HXXXY motif containing CcmE in archaea is just an artifact of historical contingency or due to functional/evolutionary constraints remains unclear.

Finally, the cyt *c* synthetase complex in bacteria contains two (CcmFH or CcmFI) or three (CcmFHI) proteins depending on the strain whereas the archaeal counterpart only contains CcmF. In *M. acetivorans*, the CcmF gene is split into two coding sequences (*ccmF1* and *ccmF2*), each of which might be derived from different bacterial groups (*Figure 8* and *Figure 8—figure supplements 5 and 6*).

Functionally important domains of the cyt *c* synthetase complex including the two histidine residues that coordinate the pseudo-periplasmic heme *b* as well as the highly conserved WWD domain are all present in CcmF2 (*Figure 2—figure supplement 1*). CcmF1 probably plays a secondary role in cyt *c* biogenesis, which might explain why a small amount of holo-cyt *c* can still be detected in the Δ*ccmF1* mutant of *M. acetivorans* (*Figure 2*). In bacteria and *Arabidopsis*, CcmH/I have been shown to interact with the apo-cyt *c* and deliver it to CcmF (*Kranz et al., 2009*; *Setterdahl et al., 2000*; *Meyer et al., 2005*). It is unlikely that *M. acetivorans* and other archaea have replaced CcmH/I with a non-orthologous protein as heterologous expression of *ccmABCEF1F2* in the non-native host, *M. barkeri* Fusaro, seems to be sufficient for cyt *c* biogenesis (*Figure 5*). However, we cannot distinguish between the possibility that *M. barkeri* Fusaro also encodes this non-orthologous protein or that the standalone CcmF in *M. acetivorans* performs the function of CcmFH in *E. coli*.

Within the Archaea, cyt *c* are especially enriched in the family *Methanosarcinaceae* wherein these proteins have likely fostered tremendous metabolic innovation (*Mand and Metcalf, 2019*). Our work and other studies (*Holmes et al., 2019*; *Ferry, 2020*; *Holmes et al., 2021*; *Zhou et al., 2021*) show that cyt *c* like MmcA are intricately coupled to growth under a variety of conditions in the marine methanogen *M. acetivorans* (*Figure 6*). Furthermore, membrane-spanning multi-heme cyt *c* have been shown to facilitate anaerobic methane oxidation by microbial consortia comprised of ANME archaea and sulfate reducing bacteria (*Wegener et al., 2015*; *McGlynn et al., 2015*; *Scheller et al., 2016*; *He et al., 2019*) and recent studies also suggest an important role for cyt *c* in facilitating anaerobic alkane metabolism in several members of the *Methanosarcinaceae* (*Borrel et al., 2019*). Since ANME and alkane oxidizing archaea are yet to be isolated in pure culture, our ability to manipulate the biogenesis of cyt *c* also renders *M. acetivorans* as an ideal platform to perform functional analyses of diverse cyt *c* across the *Methanosarcinaceae*. Curiously, even though cyt *c* are widely present in members of the *Methanosarcinaceae*, there are many extant strains, such as *M. barkeri* Fusaro, that lack the Ccm machinery altogether and do not encode any cyt *c* proteins in their genome. Our evolutionary analyses provide strong evidence that the uneven distribution of the Ccm machinery and cyt *c* in the *Methanosarcinaceae* is due to gene loss events rather than independent gene gain events (*Figure 7* and *Figure 7—figure supplement 1*). It is also worth noting that many of the extant lineages that lack the Ccm machinery and cyt *c* are found in freshwater or anthropogenic sources (like waste digestors); reciprocally, the Ccm machinery and diverse cyt *c* are especially conserved in lineages derived from marine environments (*Figure 7* and *Figure 7—figure supplement 1*). This pattern is indicative of the likelihood that there might be environment-specific selective pressure to retain or discard cyt *c* and the associated biogenesis machinery. However, the specific biotic or abiotic environmental cues that underlie this pattern remain unclear at the moment.

Previous studies (*Allen et al., 2006*; *Kletzin et al., 2015*) have used some of the unique features that unify the archaeal Ccm machinery, notably the absence of CcmDHI and a conserved CXXXY heme-binding domain in CcmE, to support an ancestral origin of the Ccm machinery in the Archaea. Our phylogenetic analyses (*Figure 8* and *Figure 8—figure supplements 1–6*) contradict this hypothesis and provide strong evidence supporting the hypothesis that there have been many independent HGT events that have cross-pollinated the Ccm machinery between the Archaea and Bacteria. In fact, even the Ccm machinery in one single archaeon, like *M. acetivorans,* may not have been derived from a single source (*Figure 8* and *Figure 8—figure supplements 1–6*). Taken together, the streamlined Ccm machinery in the Archaea is functionally uniform despite independent evolutionary origins.

## Materials and methods
### In silico design of sgRNAs for Cas9-mediated genome editing
Twenty bp target sequences for Cas9-mediated genome editing in this study are listed in *Supplementary file 3*. All target sequences were chosen using the CRISPR site finder tool in Geneious Prime version 11.0 with the following parameters: (i) the PAM site was set to NGG at the 3′ end, (ii) 0 mismatches were allowed against off-targets, and (iii) 0 mismatches were allowed to be indels. Activity scores were predicted using the methods described previously (*Doench et al., 2016*). The *M. acetivorans* chromosome and the plasmid pC2A were used to score off-target binding sites.

## Plasmid construction

All plasmids used in this study are listed in *Supplementary file 4* and the primers used to generate the plasmids are listed in *Supplementary file 5*. Plasmids for Cas9-mediated genome editing were designed using pDN201 as the backbone described previously (*Nayak and Metcalf, 2017*). Briefly, PCR fragments with the P*mtaCB1* promoter from *M. acetivorans*, single guide RNAs (sgRNA(s)) targeting gene(s) of interest, and the *mtaCB1* terminator from *M. acetivorans* were fused to *AscI* linearized pDN201 using the Gibson assembly method as described previously (*Nayak and Metcalf, 2017*). Subsequently, PCR fragments with the repair template were fused to the sgRNA containing vector linearized with *PmeI* using the Gibson assembly method as described previously (*Nayak and Metcalf, 2017*). All plasmid-based overexpression constructs were constructed using pJK029A (*Guss et al., 2008*) as the backbone. Fragments containing the gene of interest and promoter, terminators, or TAP tags as indicated in *Supplementary file 4* were amplified and fused to the P*mcrB*(*tetO4*) promoter in pJK029A (*Guss et al., 2008*) linearized with *NdeI* and *HindIII* using the Gibson assembly method as described previously (*Nayak and Metcalf, 2017*). A cointegrate of the pDN201-derived plasmid or pJK029A-derived plasmid and pAMG40, containing the pC2A origin of replication, was obtained using the Gateway BP Clonase II Enzyme Mix (Thermo Fisher Scientific, Waltham, MA) prior to transformation in *Methanosarcina* spp. Standard techniques were used for the isolation and manipulation of plasmid DNA. WM4489, a DH10ß derivative engineered to control copy-number of oriV-based plasmids (*Kim et al., 2012*), was used as the host strain for all plasmids generated in this study (*Supplementary file 4*). WM4489 was transformed by electroporation at 1.8 kV using an *E. coli* Gene Pulser (Bio-Rad, Hercules, CA). All pDN201-derived plasmids were verified by Sanger sequencing at the UC Berkeley DNA Sequencing Facility and all pAMG40 cointegrates were verified by restriction endonuclease analysis.

## Strains and growth media

All strains used in this study are listed in *Supplementary file 6*. All archaeal strains were grown in single-cell morphology (*Sowers et al., 1993*) at 37°C without shaking in bicarbonate-buffered HS liquid medium with $N_2/CO_2$ (80/20) in the headspace. For transformations and growth analyses, 10 mL cultures were grown in Balch tubes with $N_2/CO_2$ (80/20) at 55–69 kPa in the headspace. For protein purifications, 250 mL cultures were grown in anaerobic bottles with $N_2/CO_2$ (80/20) at 21–35 kPa in the headspace. All *M. barkeri* Fusaro cells were cultivated in liquid medium supplemented with 125 mM methanol. For mutant generation, *M. acetivorans* and *M. barkeri* Fusaro were plated on agar-solidified HS medium (1.6% agar w/v) with 50 mM TMA or 62.5 mM methanol as the carbon and energy substrate, respectively. Solid media plates were incubated in an intra-chamber anaerobic incubator maintained at 37°C with $N_2/CO_2/H_2S$ (79.9/20/0.1) in the headspace, as described previously (*Metcalf et al., 1998*). Puromycin (RPI, Mount Prospect, IL) to a final concentration of 2 µg/mL and the purine analog 8ADP (CarboSynth, San Diego, CA) to a final concentration of 20 µg/mL were added from sterile, anaerobic stock solutions to select for transformants containing the *pac* (puromycin transacetylase) cassette and to select against the *hpt* (phosphoribosyltransferase) cassette encoded on pC2A-based plasmids, respectively. Anaerobic, sterile stocks of tetracycline hydrochloride in deionized water were prepared fresh shortly before use and added to a final concentration of 100 µg/mL. All mutant strains were verified by Sanger sequencing at the UC Berkeley DNA Sequencing Facility. All *E. coli* strains were grown in Lysogeny broth (LB) or LB-agar at 37°C with appropriate antibiotics (25 µg/mL kanamycin and/or 10 µg/mL chloramphenicol) as indicated for different constructs (*Supplementary file 4*). All liquid cultures of *E. coli* were grown shaking at 250 rpm. For plasmid preparation, cultures were supplemented with 10 mM rhamnose to increase the plasmid copy number of the pDN201- and pJK029A-derived plasmids.

## Transformation of *Methanosarcina* spp.

Liposome-mediated transformation was used for *M. acetivorans* and *M. barkeri* Fusaro as described previously (*Metcalf et al., 1997*). Late-exponential phase culture of *M. acetivorans* (10 mL with TMA) or *M. barkeri* Fusaro (50 mL with MeOH) and 2 µg of plasmid DNA were used for each transformation. Briefly, cells were centrifuged, the supernatant was carefully decanted, and the pellet was resuspended in 1 mL bicarbonate-buffered isotonic sucrose (0.85 M) containing 100 µM cysteine. Two µg plasmid DNA and 25 µL DOTAP (*N*-[1-(2,3-dioleoyloxy)propyl]-*N*,*N*,*N*-trimethylammonium

methylsulfate) (Roche Diagnostics Deutschland GmbH, Mannheim, Germany) were added to the cell suspension and incubated for 4 hr at room temperature in an anaerobic chamber with $CO_2/H_2/N_2$ (20/4/balance) in the headspace. The mix of cells, plasmid DNA, and DOTAP was inoculated in HS medium with the appropriate substrate growth medium and incubated at 37°C for 12–16 hr prior to plating. *M. acetivorans* cells were spread on puromycin-containing agar-solidified medium using a spreader whereas *M. barkeri* Fusaro cells were plated using the top-agar method described previously (*Sowers et al., 1993*; *Boccazzi et al., 2000*).

## Whole-genome resequencing of CRISPR-edited *M. acetivorans* mutants
A 10 mL culture of DDN029 in HS + 50 mM TMA incubated at 37°C was harvested at late-exponential phase ($OD_{600} \sim 0.8$). Genomic DNA was extracted using the Qiagen blood and tissue kit (Qiagen, Hilden, Germany) and the concentration of genomic DNA was measured using a Nanodrop One Microvolume UV-Vis Spectrophotmeter (Thermo Fisher Scientific, Waltham, MA). Library preparation and Illumina sequencing was performed at the Microbial Genome Sequencing Center, Pittsburgh, PA, USA. Illumina sequencing reads were aligned to the *M. acetivorans* C2A genome and mutations were identified using Breseq version 0.35.5 (*Deatherage and Barrick, 2014*). Illumina sequencing reads for DDN029 have been deposited to the Sequencing Reads Archive (SRA) with the following BioProject accession number: PRJNA800036.

## Growth assays for *M. acetivorans*
*M. acetivorans* strains were grown in single-cell morphology (*Sowers et al., 1993*) in bicarbonate-buffered HS liquid medium containing 125 mM methanol, 50 mM TMA, 40 mM sodium acetate, or 20 mM DMS. Most substrates were added to the medium prior to sterilization. DMS was added from an anaerobic stock solution maintained at 4°C immediately prior to inoculation. For growth analyses, 10 mL cultures were grown in sealed Balch tubes with $N_2/CO_2$ (80/20) at 55–69 kPa in the headspace. Growth measurements were conducted with at least three independent biological replicates derived from colony-purified isolates using optical density readings at 600 nm ($OD_{600}$). Optical density readings were obtained from a UV-Vis Spectrophotometer (Gensys 50, Thermo Fisher Scientific, Waltham, MA) outfitted with an adjustable test tube holder that could directly take $OD_{600}$ readings from cultures in a Balch tube. A Balch tube containing 10 mL HS medium with the appropriate growth substrate was used as the 'Blank' for $OD_{600}$ measurements. For growth on methanol, TMA, and DMS, cells were acclimated to the growth substrate for a minimum of five generations prior to quantitative growth measurements. Growth measurements on acetate were performed with cells transferred from HS + TMA medium. A 1:20 dilution of mid-late-exponential phase cultures was used as the inoculum for growth analyses. Growth curves were plotted on a log base 10 scale and growth rate measurements were performed individually for each replicate by calculating the slope of the line that fits as many data points on the growth curve for a linear regression coefficient ($R^2$) $\geq 0.99$. For maximum $OD_{600}$ measurements, cells were diluted 1:10 in HS medium containing the appropriate growth substrate. Growth curve plots and statistical analyses were obtained using GraphPad Prism 9.0.0.

## Purification of membrane fraction containing TAP-tagged CcmE
Protein purification was performed with 250 mL of mid-exponential phase culture grown in HS + 50 mM TMA at 37°C without shaking under anaerobic conditions. Cells were harvested by centrifugation (6000× *g*) for 20 min at 4°C and were lysed in 10 mL of hypotonic Lysis buffer (50 mM $NaH_2PO_4$, pH = 8.0) on ice for 30 min with intermittent shaking. Sodium chloride was added from a 5 M stock solution to a final concentration of 300 mM to the cell lysate. The cell lysate was clarified by centrifugation at 10,000× *g* for 10 min at 4°C, followed by separation of soluble and membrane fractions via high-speed ultracentrifugation at 100,000× *g* for 45 min at 4°C. The membrane pellets were solubilized in 4 mL Wash buffer (50 mM $NaH_2PO_4$, 300 mM NaCl, pH = 8.0) with 1% Triton X-100 (Sigma-Aldrich, St Louis, MO). The solubilized membrane fraction was loaded on a column containing 0.5 mL Strep-Tactin Superflow plus resin (50% suspension; Qiagen, Hilden, Germany) equilibrated with 4 mL of the Wash buffer. The column was washed twice with 4 mL of Wash buffer by gravity flow and the purified protein was eluted in four fractions with 0.5 mL Elution buffer (50 mM $NaH_2PO_4$, 300 mM NaCl, 2.5 mM desthiobiotin, pH = 8.0) per fraction. The protein was concentrated using 10 kDa Amicon filter (Merck Millipore, Burlington, MA). The protein concentration in each fraction was estimated using the

Bradford reagent (Sigma-Aldrich, St Louis, MO) with BSA (bovine serum albumin) as the standard per the manufacturer's instructions.

## Heme staining and immunoblotting

Heme staining using heme peroxidase assays were performed as described previously (*Feissner et al., 2003*; *Gupta et al., 2019*). For heme staining MmcA, total cell lysate of *M. acetivorans* or *M. barkeri* Fusaro was incubated at 65°C for 4 min. For heme staining CcmE, cell lysate/enriched protein samples were not heat treated. Protein samples were resolved by 12% Mini-Protean TGX denaturing SDS-PAGE gel (Bio-Rad, Hercules, CA) and transblotted to 0.2 μm PVDF membrane (Bio-Rad, Hercules, CA) using Trans-Blot Turbo transfer system (Bio-Rad, Hercules, CA). For heme staining, the SuperSignal West Femto kit (Thermo Fisher Scientific, Waltham, MA) was used to detect the heme signal on the transblotted PVDF membrane and imaging was performed with ChemiDoc MP Imaging System (Bio-Rad, Hercules, CA). All the heme stain blots were washed with 50 mL stripping buffer (60 mM Tris pH = 7 containing 2% SDS and 7 μL/mL β-mercaptoethanol) shaking at 50 rpm for 1 hr at 50°C and confirmed for the absence of any peroxidase-based signal from heme before they were used for immunoblots. FLAG-tagged proteins were probed with immunoblotting using monoclonal anti-Flag M2-Peroxidase (HRP) antibody (Sigma-Aldrich, St Louis, MO) (1/50,000× dilution) and Immobilon Western Chemiluminescent HRP Substrate (Millipore, Burlington, MA) was used for signal detection. Imaging was performed with ChemiDoc MP Imaging System (Bio-Rad, Hercules, CA). Near-equal loading of total protein was estimated using the Bradford reagent (Sigma-Aldrich, St Louis, MO) with BSA as the standard per the manufacturer's instructions. All experiments were conducted for at least two biological replicates per strain and a representative blot has been shown in the text.

## Bioinformatics analyses

Phylogenetic trees of *Methanosarcina* spp. and other strains within the *Methanosarcinales* (as depicted in *Figure 7a* and *Figure 7—figure supplement 1*) were obtained from AnnoTree (*Mendler et al., 2019*). AnnoTree was used for functional annotation of *ccm* genes using the following KEGG annotations: *ccmA* (K02913), *ccmB* (K02194), *ccmC* (K02195), *ccmE* (K02197), *ccmF* (02198). For gene trees, 500 closest homologs were extracted from the NCBI non-redundant protein database using the corresponding Ccm gene sequence from *M. acetivorans* as the query in BLAST-P searches. The amino acid sequences of these proteins were aligned using MUSCLE 3.7 on the CIPRES Science Gateway cluster V3.3 (*Miller et al., 2010*). Maximum-likelihood trees were generated using RAxML V8.2.12 with the Jones-Taylor-Thornton substitution matrix on the CIPRES Science Gateway cluster V3.3 (*Miller et al., 2010*). Trees were displayed using Fig Tree V1.4.3 (http://tree.bio.ed.ac.uk/software/figtree/). Gene ortholog neighborhood was obtained using the bidirectional best hits for the corresponding *ccm* gene using the Integrated Microbial Genomes and Microbiomes platform containing annotated isolate genome and metagenome datasets sequenced at the Joint Genome Institute (*Chen et al., 2019*).

## Acknowledgements

The authors would like to acknowledge members of the Nayak lab for their feedback and input. The authors acknowledge funding from the 'New Tools for Advancing Model Systems in Aquatic Symbiosis' program from the Gordon and Betty Moore Foundation (GBMF#9324) to DDN and DG for genetic analyses of cyt *c* biogenesis in *M. acetivorans* and the Simons Early Career Investigator in Marine Microbial Ecology and Evolution Award to DDN and KES for studies investigating the role of CcmE in cyt *c* biogenesis. This research was also funded by the Searle Scholars Program sponsored by the Kinship Foundation (to DDN), the Rose Hills Innovator Grant (to DDN), the Beckman Young Investigator Award sponsored by the Arnold and Mabel Beckman Foundation (to DDN), the Packard Fellowship in Science and Engineering sponsored by the David and Lucille Packard Foundation (to DDN), funding from the Shurl and Kay Curci Foundation (to DDN), and startup funds from the Department of Molecular and Cell Biology at UC Berkeley (to DDN). The funders had no role in the conceptualization and writing of this manuscript or the decision to submit the work for publication.

## Additional information

### Funding

| Funder | Grant reference number | Author |
|---|---|---|
| Gordon and Betty Moore Foundation | GBMF#9324 | Dipti D Nayak<br>Dinesh Gupta |
| Simons Foundation | Simons Early Career Investigator in Marine Microbial Ecology and Evolution Award (822981) | Dipti D Nayak<br>Katie E Shalvarjian |
| David and Lucile Packard Foundation | Packard Fellowships for Science and Engineering | Dipti D Nayak |
| Searle Scholars Program | | Dipti D Nayak |
| Arnold and Mabel Beckman Foundation | Beckman Young Investigator Program | Dipti D Nayak |
| Shurl and Kay Curci Foundation | | Dipti D Nayak |

The funders had no role in study design, data collection and interpretation, or the decision to submit the work for publication.

### Author contributions

Dinesh Gupta, Conceptualization, Data curation, Formal analysis, Investigation, Methodology, Validation, Visualization, Writing - original draft, Writing - review and editing; Katie E Shalvarjian, Data curation, Formal analysis, Methodology, Validation, Writing - review and editing; Dipti D Nayak, Conceptualization, Data curation, Formal analysis, Funding acquisition, Investigation, Methodology, Project administration, Resources, Supervision, Validation, Visualization, Writing - original draft, Writing - review and editing

### Author ORCIDs

Dinesh Gupta (ID) http://orcid.org/0000-0001-9108-9669
Katie E Shalvarjian (ID) http://orcid.org/0000-0002-4792-8477
Dipti D Nayak (ID) http://orcid.org/0000-0002-3449-3419

### Decision letter and Author response

Decision letter https://doi.org/10.7554/eLife.76970.sa1
Author response https://doi.org/10.7554/eLife.76970.sa2

## Additional files

### Supplementary files

• Supplementary file 1. List of mutations in CRISPR-edited mutant strain DDN029 containing a ΔccmABC in-frame deletion mutation.

• Supplementary file 2. Growth data of *Methanosarcina acetivorans* strains shown in *Figure 6*.

• Supplementary file 3. List of target sequences used in this study.

• Supplementary file 4. List of plasmids used in this study.

• Supplementary file 5. List of primers used in this study.

• Supplementary file 6. List of *Methanosarcina* strains used in this study.

• Transparent reporting form

### Data availability

All data generated or analysed during this study are included in the manuscript and supporting file; sequencing data have been deposited in the NCBI SRA (Sequence Read Archive) under bioproject number PRJNA800036. Source data files for Figure1b, Figure 2c, Figure 3d, Figure 4c and Figure 3—figure supplement 1, Figure 4—figure supplement 1 and 2, are provided.

The following dataset was generated:

| Author(s) | Year | Dataset title | Dataset URL | Database and Identifier |
|---|---|---|---|---|
| Nayak DD, Gupta D, Shalvarjian KE | 2022 | Whole genome re-sequencing data for the *Methanosarcina acetivorans* ccmABC deletion mutant | https://www.ncbi.nlm.nih.gov/sra/?term=PRJNA800036 | NCBI Sequence Read Archive, PRJNA800036 |

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
