## [Editor Report]

Within this manuscript the authors set out to determine the biogenesis of c-type cytochromes in methane producing archaea called methanogens. Compared to the bacterical cytochrome c assembly system, genes like ccmD, ccmH and ccmI are not found in archaea that contain a functional cytochrome Cs. They show that the proteins encoded by the ccmABCEF genes of *Methanosarcina acetivorans* are both essential and sufficient for cyt c biogenesis. They also show the substrate specific role of the mmcA y cyt c *M. acetivorans*. The authors do this using a combination of genetic, molecular, and physiological and biochemical analyses. These data are of high interest of scientists working on bioenergetics and yt c biogenesis in archaea.

---

## [Decision Letter]

**Decision letter after peer review:**

Thank you for submitting your article "An Archaea-Specific c-type Cytochrome Maturation Machinery is Crucial for Methanogenesis in *Methanosarcina acetivorans*" for consideration by *eLife*. Your article has been reviewed by 2 peer reviewers, including Sonja V Albers as the Reviewing Editor and Reviewer #1, and the evaluation has been overseen by Gisela Storz as the Senior Editor. The following individual involved in the review of your submission has agreed to reveal their identity: Cornelia Welte (Reviewer #2).

Essential revisions:

1) Please describe the number of biological and technical repetitions that were performed for each experiment (also for the western blots).

2) L107: the I in DIET stands for 'interspecies', not 'interdomain'. While in the specific case mentioned here, it is an interdomain electron transfer, in general it more broadly categorizes interspecies electron transfer. Please correct.

3) L149-150: this sentence is misleading, as it infers that MmcA is expressed in most methane-metabolizing archaea, which is incorrect. It is present only in those few methanogens that harbor the Rnf, and I'm not even sure that the ANME Rnf's contain it. Please rephrase the sentence to better reflect this.

4) L170: I was confused here by the +Control in Figure 2c. The authors mean plus empty vector? Can you please specify this in the legend?

5) L202-204: I think the authors need to explain this better, as this statement doesn't match their Figure in my view. In the figure, CcmD is required for dissociation of CcmE, not for the loading. Furthermore, with their essay they do not check for dissociated CcmE, only for heme loading of CcmE.

6) L215: There is a double band in the anti Flag Western Blot. Do the authors have any idea why there are two bands?

7) L269: replace free energy yield by Gibb's free energy change, as that's the correct term

8) L269: I don't think that the -164.5 kJ/mol is correct. How did the authors determine this number? According to Watkins et al. 2012 doi: 10.1128/AEM.03076-13 it's -172 kJ per four TMA, so per one TMA it would be only -43 kJ, whereas if you calculate it per methane, it's -57 kJ.

L286-287: I think this sentence is not so understandable – what was reduced by what? Please rephrase.

9) L321: I'm not so sure. Whereas for all methanogens, full genome sequences with one contig exists, for Methanoperedens (and other ANME), all genomes are fragmented, making gene absence analyses untrustworthy. At least you have to mention that the ANME genomes are, in contrast to all methanogens, incomplete metagenome-assembled genomes that make gene absence analyses questionable.

10) L402: this is much more strongly phrased in the Results section. I recommend to tone it down also there to be more in line, as I agree that you cannot rule out that other genes required for cyt c maturation are maybe still present in the Fusaro strain

11) L422: Methanoperedens as one of the archaea with most cyt c is a freshwater species, so I'm not so sure whether the marine environment is the most relevant driver

12) L469 and elsewhere: could the authors please use SI units instead of psi? bar?

13) L535: I really hope Methanosarcina was cultured anaerobically, not aerobically

---

## [Author Response]

Essential revisions:1) Please describe the number of biological and technical repetitions that were performed for each experiment (also for the western blots).

We have included these details for each relevant experiment in the Materials and methods section; please see lines 513-514 and lines 556-557 in the revised manuscript. We performed growth curves with at least three biological replicates.

Western blots and heme stains were conducted with at least two biological replicates per strain. Given the constraints posed by the number of lanes in a protein gel, and the need to accommodate all strains in a single gel, we were unable to accommodate replicates in Figures 1-5. That said, the data shown in the figures are representative of the replicate experiments performed.

2) L107: the I in DIET stands for 'interspecies', not 'interdomain'. While in the specific case mentioned here, it is an interdomain electron transfer, in general it more broadly categorizes interspecies electron transfer. Please correct.

We have corrected this typo. Please see line 92 in the revised manuscript.

3) L149-150: this sentence is misleading, as it infers that MmcA is expressed in most methane-metabolizing archaea, which is incorrect. It is present only in those few methanogens that harbor the Rnf, and I'm not even sure that the ANME Rnf's contain it. Please rephrase the sentence to better reflect this.

We agree with the reviewer and have modified the sentence to clarify and rephrase the statement; please see lines 134-135. Based on a recent study (https://journals.plos.org/plosbiology/article?id=10.1371/journal.pbio.3001508), it does look like the MAGs from ANME archaea that contain Rnf also encode MmcA.

4) L170: I was confused here by the +Control in Figure 2c. The authors mean plus empty vector? Can you please specify this in the legend?

Yes, we have clarified this in the figure legend. Please see lines 812 and 869 in the revised manuscript and we have also modified the legend accompanying the figure accordingly.

5) L202-204: I think the authors need to explain this better, as this statement doesn't match their Figure in my view. In the figure, CcmD is required for dissociation of CcmE, not for the loading. Furthermore, with their essay they do not check for dissociated CcmE, only for heme loading of CcmE.

We recognize that our assay did not directly test for the dissociation of CcmE. However, dissociation is required for heme-bound CcmE to transfer the heme group to MmcA (as shown in Figure 4—figure supplement 2), which allows us to conclude that heme-bound CcmE has indeed dissociated from CcmABC. Since our experiments did not directly test dissociation of CcmE, aspointed out by the reviewer, we have softened our claim and conclusions. Please see lines 188-189.

6) L215: There is a double band in the anti Flag Western Blot. Do the authors have any idea why there are two bands?

In figure 4c, we notice only band with heme staining (top panel) and two bands in the anti-flag western blot (bottom panel). It is likely that the second band is a degradation product of the tagged version of CcmE that does not bind heme. We do not notice any cross-reactivity with the anti-FLAG antibody in the strain expressing the C120H mutant of CcmE (which likely does not produce a stable protein), which further corroborates our hypothesis that the second band in the anti-FLAG western is a CcmE degradation product rather than any other protein in the cell.

7) L269: replace free energy yield by Gibb's free energy change, as that's the correct term

We have made the edits per the reviewer’s suggestion. Please see line 254 in the revised manuscript.

8) L269: I don't think that the -164.5 kJ/mol is correct. How did the authors determine this number? According to Watkins et al. 2012 doi: 10.1128/AEM.03076-13 it's -172 kJ per four TMA, so per one TMA it would be only -43 kJ, whereas if you calculate it per methane, it's -57 kJ.

We appreciate the reviewers concern but would like to point out that the number stated in the manuscript is correct per the calculations made based on data provided by Thauer, Jungermann, and Decker,1977 (https://journals.asm.org/doi/pdf/10.1128/br.41.1.100-180.1977), which is considered the gold standard in the field.

For further corroboration re: the ∆G°’ during methanogenic growth on methylated amines, we would also like to point out two additional papers: https://link.springer.com/content/pdf/10.1007/BF02602436.pdf (see table 1) and https://journals.asm.org/doi/full/10.1128/JB.00535-06 (see table 4), which are both in agreement with numbers stated in our manuscript.

L286-287: I think this sentence is not so understandable – what was reduced by what? Please rephrase.

We have rephrased these sentences per the reviewer’s request. Please see lines 266-272 in the revised manuscript.

9) L321: I'm not so sure. Whereas for all methanogens, full genome sequences with one contig exists, for Methanoperedens (and other ANME), all genomes are fragmented, making gene absence analyses untrustworthy. At least you have to mention that the ANME genomes are, in contrast to all methanogens, incomplete metagenome-assembled genomes that make gene absence analyses questionable.

Yes, absolutely. We have changed the phrasing of our results to only make conclusive comments about gene absence in methanogenic isolates with completely sequences genomes. Please see line 304-309 in the revised manuscript.

10) L402: this is much more strongly phrased in the Results section. I recommend to tone it down also there to be more in line, as I agree that you cannot rule out that other genes required for cyt c maturation are maybe still present in the Fusaro strain

Per the reviewer’s request we have added a sentence in the discussion to indicate that we cannot rule out the possibility that some genes for cyt c maturation are still present in *M. barkeri*. Please see lines 387-389 in the revised manuscript.

11) L422: Methanoperedens as one of the archaea with most cyt c is a freshwater species, so I'm not so sure whether the marine environment is the most relevant driver

Yes, absolutely. We have toned down our claim to address the reviewer’s comment. Please see line 409 in the revised manuscript.

12) L469 and elsewhere: could the authors please use SI units instead of psi? bar?

We have converted all pressure units to KPa. Please see lines 455, 456 and 503 in the revised manuscript.

13) L535: I really hope Methanosarcina was cultured anaerobically, not aerobically

Indeed, we have fixed this typo. Please see line 522 in the revised manuscript.